# Beyond Testers' Biases:
# Guiding Model Testing with Knowledge Bases using LLMs

**Chenyang Yang**[1]     **Rishabh Rustogi**[1]     **Rachel Brower-Sinning**[2]     **Grace A. Lewis**[2]
**Christian Kästner**[1]     **Tongshuang Wu**[1]
[1]Carnegie Mellon University     [2]Carnegie Mellon Software Engineering Institute

## Abstract

Current model testing work has mostly focused on creating test cases. Identifying what to test is a step that is largely ignored and poorly supported. We propose WEAVER, an interactive tool that supports requirements elicitation for guiding model testing.[1] WEAVER uses large language models to generate knowledge bases and recommends concepts from them interactively, allowing testers to elicit requirements for further testing. WEAVER provides rich external knowledge to testers, and encourages testers to systematically explore diverse concepts beyond their own biases. In a user study, we show that both NLP experts and non-experts identified more, as well as more diverse concepts worth testing when using WEAVER. Collectively, they found more than 200 failing test cases for stance detection with zero-shot ChatGPT. Our case studies further show that WEAVER can help practitioners test models in real-world settings, where developers define more nuanced application scenarios (e.g., code understanding and transcript summarization) using LLMs.

## 1 Introduction

Despite being increasingly deployed in real-world products, ML models still suffer from false-hoods (Maynez et al., 2020), biases (Shah et al., 2020), and shortcuts (Geirhos et al., 2020), leading to usability, fairness, and safety issues in those products (Liang et al., 2023; Nahar et al., 2023). For example, toxicity detection models are used by social media platforms to flag or remove harmful content, but their biases amplify harm against minority groups (Sap et al., 2019). As standard benchmarks are often too coarse to expose these issues, recent work has proposed to test nuanced behaviors of ML models (Ribeiro et al., 2020; Goel et al., 2021; Ribeiro and Lundberg, 2022).

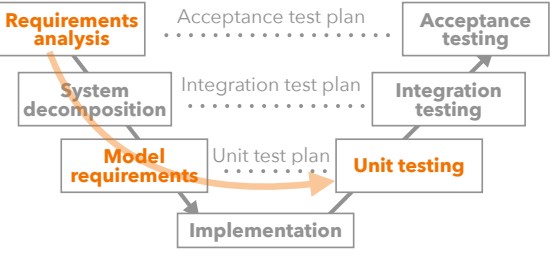

Figure 1: The V-model (Sommerville, 2015), a widely used development process in many fields of engineering, adapted for testing models within a system. This layering and planning is also compatible with more agile development approaches, where the different activities may be iterated in different ways but are still linked.

Such model testing of nuanced behaviors usually requires *translating behavior expectations into test cases (input-output pairs).* To enable such test case creation, prior work has taken inspiration from the long-established *software testing* approaches: For example, in their CheckList framework, Ribeiro et al. (2020) used *templates* to form minimal functionality test cases, which was inspired by *unit tests.* Morris et al. (2020)'s work on editing inputs (e.g., synonym swap) for testing model invariances is akin to *metamorphic testing.* To enable testing generative models, Ribeiro (2023) have also explored specifying properties that any correct output should follow, similar to *property-based testing.*

However, prior work has focused on *how to write test cases,* not *what tests to write.* In software engineering, tests are fundamentally grounded in requirements and design work, as commonly expressed in the V-model (Sommerville, 2015, Figure 1). Ideally, each test can be traced back to a requirement and all requirements are tested. Software engineering research has long established the importance of requirements for development *and* testing, and studied many approaches for *requirements elicitation* (Van Lamsweerde, 2009).

In comparison, little work has explicitly supported identifying *what to test* in model testing. Researchers and practitioners seem to rely mostly

---

[1]WEAVER is available open-source at https://github.com/malusamayo/Weaver.

on intuition and generic domain knowledge (Mc-Coy et al., 2019; Dhole et al., 2021), or debug a small set of issues initially identified through *error analysis* (Naik et al., 2018; Wu et al., 2019b). Such approaches are often shaped heavily by individual knowledge and biases (Rastogi et al., 2023; Lam et al., 2023), driving practitioners to focus on local areas of a few related concepts where they find problems, while neglecting the larger space (Ribeiro and Lundberg, 2022), as exemplified in Figure 2. For example, to test toxicity detection models, practitioners may identify and test (1) *racism* with handcrafted test cases and (2) *robustness to paraphrasing* with CheckList. However, they are likely to miss many concepts in the space (e.g., *spreading misinformation* as a way to spread toxicity) if they have never seen or worked on similar aspects, as often observed in the fairness literature (Holstein et al., 2019).

In this work, we contribute **the concept and method of requirements engineering** (*"what to test"*) for **improving model testing**. We connect testing to requirements with WEAVER, *an interactive tool that supports requirements elicitation for guiding model testing.* Our goal is to provide comprehensive external knowledge, encouraging testers to systematically explore diverse concepts beyond their biases (Figure 2), while balancing the completeness of requirements collection and the effort of practitioners exploring these requirements. WEAVER systematically generates knowledge bases (KB) by querying large language models (LLMs) with well-defined relations from ConceptNet, allowing testers to elicit requirements *for any use cases* (as opposed to being limited by pre-built KBs). For example, in Figure 3, from a *seed concept* "online toxicity", the LLM generates a KB showing "spreading misinformation" with relation "done via." To not overwhelm users and encourage iterative exploration, WEAVER shows only a subset of the concepts from the KBs, balancing relevance and diversity, while allowing users to still explore more concepts on demand.

We demonstrate the usefulness and generality of WEAVER through a user study and case studies. In the user study (§4), users identified more, as well as more diverse concepts worth testing when using WEAVER. Collectively, they found more than 200 failing test cases for stance detection with zero-shot ChatGPT (gpt-turbo-3.5). Our case studies (§5) further show that WEAVER can help practitioners

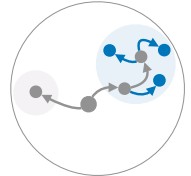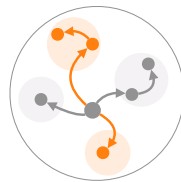

Figure 2: Local exploitation (left) vs. global exploration (right). Most model testing is opportunistic in terms of *what to test,* often "hill-climbing" by searching near existing problems, with the risk of getting stuck in local areas. In contrast, WEAVER supports exploring concepts globally across the entire space.

test models in real-world settings, from transcript summarization to code understanding.

## 2 WEAVER

Our key idea is to provide a knowledge base to support *systematic* exploration of domain concepts to guide model testing. This allows testers to consider requirements broadly, mitigating their biases as testers otherwise tend to opportunistically explore concepts in local areas (Figure 2). Our tool, WEAVER, has three primary building blocks: (1) An LLM-generated knowledge base for a given testing task, for encouraging more *systematic* and diverse requirement elicitation beyond individual biases; (2) a graph-inspired recommendation paradigm that prioritizes *diverse yet relevant* concepts for user inspection; and (3) an intuitive interface that can be easily paired with any test case creation methods, for supporting users to *interactively navigate* the knowledge base(s) for their own purposes. Below, we walk through the design choices and corresponding rationales.

### 2.1 LLM-generated Knowledge Base

To support testers to explore different concepts relevant to the problem, WEAVER needs to **provide comprehensive knowledge beyond individual testers' biases.** As such, we choose to power WEAVER using *knowledge bases (KBs) generated by LLMs.* As LLMs store diverse world knowledge that can be easily extracted through prompting (Cohen et al., 2023), they empower WEAVER to support a wide range of domains, tasks, and topics.

We start knowledge base construction with a *seed concept*, i.e., a user-provided high-level term that can represent their tasks well. These seeds can be as simple as the task name and description, or can be more customized depending on user needs. Using the seed, we will then automatically query an

LLM[2] to build a partial knowledge base of related concepts (Wang et al., 2020; Cohen et al., 2023). Specifically, we iteratively prompt LLMs for entities or concepts that have different relations to the queried concept, using fluent zero-shot prompts paraphrased from well-established relations used in knowledge bases. For example, prompting LLMs with "*List some types of online toxicity.*" can help us extract specific *TypeOf* online toxicity. By default, we use 25 relations from ConceptNet[3] (Speer et al., 2017), e.g., *MotivatedBy* and *LocatedAt*, and manually curated corresponding zero-shot templates. These relations prove to be reusable across many different domains in our studies, but users can also specify custom relations they want to explore (a complete list of prompts in Appendix A).

As the KB is used to support exploration (explained more below), we initially pre-generate two layers of the KB, and iteratively expand the knowledge base based on user interactions.

## 2.2 Recommend Diverse & Relevant Concepts

A comprehensive KB comes at a higher cost of exploration—a single tester can easily get overwhelmed if presented with the entire KB. To assist users to navigate through the KB more efficiently, we employ the "overview first, details-on-demand" taxonomy (Shneiderman, 1996): We provide initial recommendations as starting points to user explorations. In particular, we strive to **make recommendations that are diverse** (so as to bring diverse perspectives to users) **but still relevant** (so users do not feel disconnected from their goals).

The trade-off between *relevant* and *diverse* resonates with the common exploration-exploitation trade-off in information retrieval (Athukorala et al., 2016), and can be naturally translated to a graph problem: If we have all candidate concepts to form a fully-connected graph, where edge weights are distances between different concepts and node weights are the concepts' relevance to the queried concept, then our goal becomes to recommend a *diverse* subset where concepts have large distances

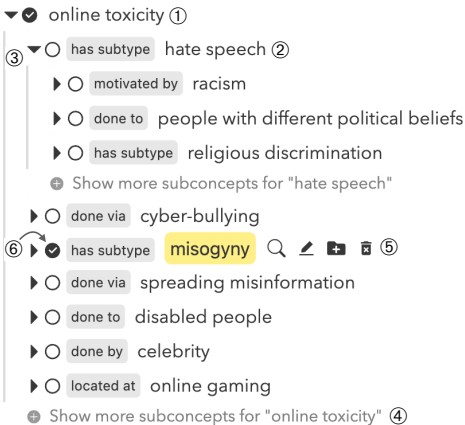

Figure 3: WEAVER interface, where users can interactively explore concepts in the LLM-generated KB to elicit requirements for model testing. ①: seed concept, ②: recommended children concepts with relations (translated to a user-friendly form), ③: options for users to expand a concept or ④: get more recommendations on any concepts, ⑤: editing, creating, or removing concepts manually, ⑥: selecting important concepts for testing.

between each other, while they are still *relevant* in the sense that they occur frequently in the context of the queried concept. Essentially, we aim to find a subgraph $G'$ of size $k$ that maximizes a weighted ($\alpha > 0$) sum of edge weights $w_E$ (diversity) and node weights $w_V$ (relevance):

$$\arg\max_{G' \subset G, |G'|=k} w_E(G') + \alpha \cdot w_V(G')$$

To build the graph, we measure concept differences with cosine distance between concept embeddings using SentenceBERT (Reimers and Gurevych, 2019), and measure relevance with the perplexity of sentence "*{concept} often occurs in the context of {queried_concept}*", using GPT-2 (Radford et al., 2019). Since finding the optimal subgraph is computationally expensive, we apply the classic greedy peeling algorithm (Wormald, 1995) to approximate it in linear time. That is, we greedily remove nodes with the smallest weights (sum of node and all edge weights) one at a time until the graph size reaches $k$ (= 10 for initial recommendation, but grows with user expansion). We empirically show that the recommended concepts are of high quality and diverse in our evaluation.

## 2.3 Interactive Interface for Exploration

Besides the recommended starting points, we allow users to **iteratively and interactively locate their concepts of interest.** WEAVER visualizes the knowledge base in a tree structure (Figure 3), a representation that is familiar to most ML practitioners.

---

[2]We used OpenAI's `text-davinci-003` (Ouyang et al., 2022), but LLMs with similar instruction following capabilities should all be useful (see additional experiments on `llama-2-13b-chat` in Appendix F).

[3] We used ConceptNet relations because they represent generic semantic relations, which we expect to be more or less generalizable—an assumption that is validated by our user study and case studies. In contrast, alternative KBs (e.g., WikiData, DBpedia) tend to focus on more specific types of semantic relations that are biased towards certain domains.

The knowledge base starts with ① the seed concept users specify, with each recommended concept represented as ② a node in the tree, accompanied by its relation to the parent concept. The interface allows users to identify concepts by ③ diving deeper and ④ exploring broadly before ⑥ selecting a concept to test. Alternatively, users can also distill personal knowledge by ⑤ creating concepts manually.

To assist users in creating concrete test cases, WEAVER incorporates AdaTest (Ribeiro and Lundberg, 2022) as the default test case creation method, which uses LLMs to suggest test cases. However, the design of WEAVER is compatible with any other techniques to test models once requirements are identified (e.g., Zeno, Cabrera et al., 2023). The full interface including the AdaTest integration can be seen in Appendix B.

## 3 Intrinsic Evaluation

As the primary goal of WEAVER is to provide external knowledge to guide testing, it is important that the knowledge provided is comprehensive in the first place. Here, we quantitatively evaluate:

Q.1 *How comprehensive are the knowledge bases generated by* WEAVER*?*

**Tasks, data, and metrics.** We select four tasks for the evaluation: Hateful meme detection, Pedestrian detection, Stance detection for feminism, and Stance detection for climate change (task descriptions in Appendix C). These tasks cover diverse domains and modalities, and importantly, provide us with gold concepts that can be used to evaluate our LLM-generated KB. The first two tasks have been studied in prior work, and we directly use their ground-truth concepts collected from existing documents (Barzamini et al., 2022b) and user studies (Lam et al., 2023). For the last two tasks, we aggregate all concepts identified by 20 participants without using WEAVER as part of our user study (discussed later in §4), which we consider as our ground truth. Intuitively, such aggregation should help represent what concepts are generally deemed important. As shown in Table 1, the tasks have on average 144 ground-truth concepts.[4]

Independently, we generated a knowledge base for each task using WEAVER with default relations. We derived the seed concepts directly from the task names: (1) *"hateful meme"*, (2) *"pedestrian"*, (3) *"feminism"*, and (4) *"climate change."*

---

[4] All ground-truth concepts are shared at https://figshare.com/s/481a69fa1b36dbd76088.

We evaluate the comprehensiveness of the generated knowledge using *recall*, i.e., the fraction of existing concepts that also appear in the KB. Since there are many phrasing variations of the same concept, we decide that a concept is in the KB if it appears exactly the same in the KB, or our manual check decides that it matches one of the 10 most similar concepts from the KB, as measured by the cosine distance (cf. §2.1). We established that the manual process is reliable by evaluating inter-rater reliability where two authors independently labeled a random sample of 50 concepts, finding substantial agreement ($\kappa = 69.4\%$).

We also evaluate the validity of the generated knowledge using *precision*, i.e., the fraction of KB edges that are valid. Note that because our ground truths are incomplete by nature (collected from dataset analysis and user study), KB edges that are not in the ground truths can still be valid. Following prior work (Cohen et al., 2023), we performed manual validation on sampled KB edges. We sampled 50 edges from each of the four generated KBs.

**Results** Overall, our KBs cover 91% of ground-truth concepts on average (Table 1), with 81% of sampled generated edges being valid. Qualitatively, we found that there are two distinct types of concepts the KB failed to cover: First, there are some very specific concepts (e.g., *old photo* in hateful meme detection). Although the 2-layer KB does not cover them, it does often cover their hypernyms (e.g., *photo*). Therefore, these concepts can be discovered if users choose to explore deeper. Second, some concepts are interactions of two concepts (e.g., *fossil fuel uses in developing countries* in climate change stance detection). These can be identified by users manually, as both of their components (*fossil fuel uses* and *developing countries*) usually already exist in the KB.

## 4 User Study

Does WEAVER support effective requirements elicitation? We conduct a user study to evaluate:

Q.2 *To what degree does* WEAVER *help users explore concepts faster?*

Q.3 *To what degree does* WEAVER *help users explore concepts broadly?*

Q.4 *How much does* WEAVER *mitigate user biases when exploring concepts?*

We expect that our interaction design (§2.3) supports faster exploration (Q.2) and that the recommendations (§2.2) support broader and less biased

| Task | Recall | Precision | # Concept |
|------|--------|-----------|-----------|
| Hateful meme detection | 93.1% | 88.0% | 101 |
| Pedestrian detection | 91.8% | 74.0% | 146 |
| Stance detection for feminism | 86.9% | 84.0% | 145 |
| Stance det. for climate change | 91.4% | 76.0% | 185 |
| Average | 90.6% | 80.5% | 144 |

Table 1: Knowledge bases generated by WEAVER cover 90.6% of existing concepts on average.

exploration (Q.3 and Q.4).

### 4.1 Study Design

**Conditions.** We design an IRB-approved user study as a *within-subject controlled experiment,* where participants test models in two conditions: A *treatment condition*, where users use WEAVER to find concepts for testing, and a *control condition*, where users add the concepts manually while they explore test cases. In both conditions, users have access to AdaTest's LLM-based test case suggestions (cf. §2.3). In essence, the *control* interface is a re-implementation of AdaTest with WEAVER's interface and interaction experience.

**Tasks and models.** We select two tasks of similar difficulty for our user study: Stance detection for feminism, and stance detection for climate change. They are accessible to participants from different backgrounds. We had participants test the performance of zero-shot ChatGPT (OpenAI, 2022) for both tasks, as we observed that it easily outperformed any available fine-tuned models on Huggingface—the latter failed at simple test cases (full prompts in Appendix A).

**Procedure.** We recruited 20 participants (graduate students with varying ML/NLP experience, details in Appendix D.1) for a 90-minute experiment session. We started by walking through the study instructions and asked them to try WEAVER in an interactive tutorial session. Then participants tested the two aforementioned stance detection models for 30 minutes each, one in the *treatment* condition and one in *control* condition. To mitigate learning effects, we use a Latin square design (Box, 2009) with four groups, counterbalancing (1) which condition a participant encounters first, and (2) which model is tested first. Within each session, they were first asked to perform model testing for 25 minutes, and then identify (select or create) concepts worth future testing for 5 minutes. The first phase

of model testing is designed to ground participants in what concepts are worth testing. The second phase of concept exploration is designed to approximate a longer time of model testing. This final design was derived from an earlier pilot study, where we observed that writing test cases for each concept took more time than identifying interesting concepts (WEAVER's objective). In the end, participants filled out a post-study survey (details in Appendix D.2). Participants were compensated for their time.

**Metrics and analysis.** We use two measurements to approximate participants' exploration procedure: (1) the number of concepts they explore (representing exploration speed, Q.2), and (2) the number of *distinct* concepts they explore (Q.3).

Specifically, for *distinctiveness*, we want to distinguish the local vs. global exploration patterns (cf. Figure 2), which requires us to locate *clusters* of similar concepts, or concepts that only differ in granularity. Quantitative, this is reflected through inter-relevance between concepts, e.g., *rising sea level* should be considered close to *sea surface temperature increase* but distinct from *waste management*. To find a set of distinct concept clusters, we again measure the concept distance using SentenceBERT, and run Hierarchical Clustering (Ward Jr, 1963) on all available concepts collectively selected or created by our 20 user study participants, which, as argued in §3, forms a representative set of what end users may care about for a given task. Note that we do not use all concepts from our KB for clustering as it would influence the ground truth. Hierarchical clustering allows us to choose concept clusters that have similar granularities using a single distance threshold. Empirically, we use the threshold of 0.7, which produces reasonably distinct clusters for both tasks (41 and 46 clusters for *feminism* and *climate change* respectively with an average size of 6.1 concepts).

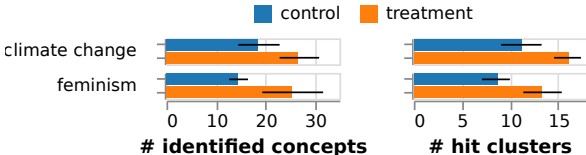

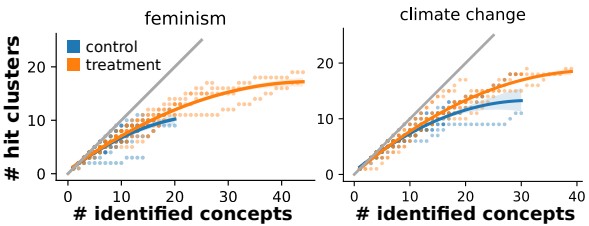

Figure 4: Participants with WEAVER identified more concepts and hit more clusters.

Figure 5: Participants without WEAVER converged to hitting previously identified concept clusters as they identified more concepts. The gap between the two groups widened over the process.

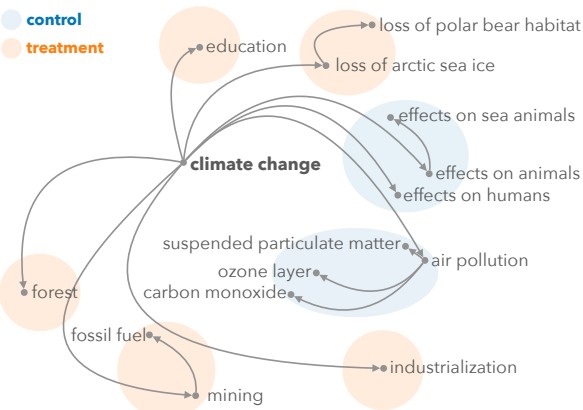

Figure 6: We project the SentenceBERT embeddings of concepts explored by two participants (P15 and P16) into a 2D space, using t-SNE (van der Maaten and Hinton, 2008). Without WEAVER, participants explore less space, as they performed more local exploitation than global exploration.

As such, *distinctiveness* is represented by the *number of hit cluster* in each user's exploration.

We analyze both measurements with a repeated measures ANOVA analysis, which highlights the significance of our test condition (whether participants use WEAVER) while considering the potential impact from other independent variables. In our analysis, we test to what degree our tool, the task, and the order (tool first or tool last) explain variance in the outcome variables. Detailed analysis results can be found in Appendix D.3.

## 4.2 Results

**WEAVER helps users identify more concepts (Q.2).** We first observe that with WEAVER, participants identified 57.6% ($p < 0.01$) more concepts (Figure 4). This is likely because users can more easily explore a wider range of concepts with the external knowledge base, as confirmed by participant P6: *"... (KB) gives ideas beyond what I had in mind so I explored a wider base of concepts."*

We also observe that bug-finding is relatively independent of concept exploration. On average, participants found around 11 failing test cases (0.44 per minute), regardless of whether they used WEAVER or not. Since testing is orthogonal, participants who explore more concepts will find more failing test cases. We expect that if participants test longer, those with WEAVER will find more bugs while others will run out of concepts to test.

**WEAVER helps users cover more concept clusters (Q.3).** More interestingly, we observe that with WEAVER, participants not only found more concepts but also covered 47.7% ($p < 0.01$) more clusters in the space, i.e., they also explored more diverse concepts (Figure 4). This aligns with the survey responses, where 80% of participants agree that WEAVER helps them test the model more holistically and 76% of participants agree that WEAVER helps them find more diverse model bugs. We conjecture that this is because users with WEAVER explore more concepts not only in quantity (Q.2) but also in diversity, which is confirmed by many participants, e.g., *"... (KB) encouraged me to explore more areas of the domain than I would have otherwise considered"* (P10).

Looking at their exploration trajectory (Figure 5), we see evidence indicating that WEAVER enables users to *continuously discover distinct concepts*. In contrast, participants in the *control* condition converged to hitting previously identified concept clusters as they identified more concepts. We observed that without WEAVER, participants tended to refine existing concepts later in the exploration, as exploring new distinct areas becomes increasingly difficult.

These contrasting trajectories eventually lead to different exploration results. As reflected in Figure 6, the participant without WEAVER performed noticeably more local exploitation, finding highly related concepts in local areas, whereas WEAVER helped the other participant explore more diverse

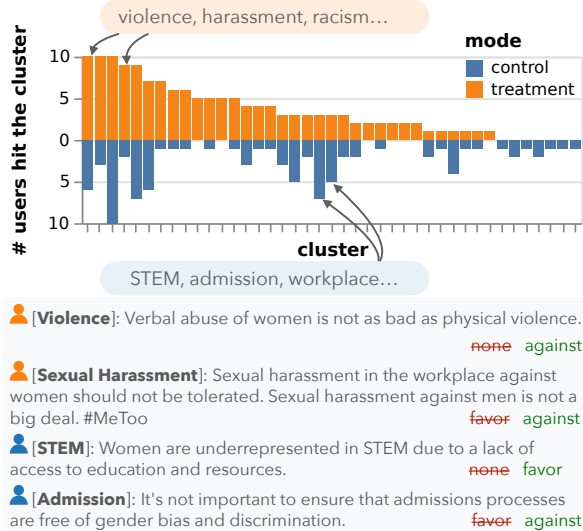

Figure 7: Visualizing the identified concepts for *feminism* by their corresponding clusters, we see participants in different conditions had different focuses in model testing. This shows that WEAVER suggests concepts complementary to human intuitions. In each test, the red strike-through labels are the wrong model predictions and the green ones are user-specified ground-truths.

concepts globally, without losing the ability to dive deeper into a few local areas.

**WEAVER shifts users towards a different concept distribution (Q.4).** We also observe that participants collectively explored different concepts with WEAVER, as shown in Figure 7. Some concepts (e.g., *violence*) are much more explored by participants with WEAVER. This suggests that WEAVER helps mitigate participants' own biases and explore new concepts, as supported by participant P13: *"... (KB) gives some inspiration in concepts I would not have otherwise thought of..."*

That said, WEAVER also brings in its own biases, e.g. participants with WEAVER rarely explored concepts like *STEM* compared to those without, possibly because they were too heavily anchored by the suggested concepts. This indicates that humans and knowledge bases are complementary – future work can support humans to better exploit their prior knowledge while exploring diverse concepts.

## 5 Case Studies

Using two case studies, we demonstrate that WEAVER can help practitioners test their own models and find various bugs in real-world settings, and has the potential to provide support beyond post-hoc model testing. In the studies, we provided sufficient supports to the practitioners, including

integrating WEAVER into their natural evaluation environment (Jupyter Notebook), joining each user to explore their models for approximately three hours, and offering feedback and discussions whenever necessary (e.g., as practitioners brainstormed their seed concepts).

**Case selection.** We approached researchers in our contact network to identify projects that were actively developing and improving models to be integrated into software products, such that (1) model testing has real stakes for them, and (2) the model needs to meet real-world requirements of a product, not just succeed on a standard benchmark.

We ended up recruiting practitioners working on two projects that matched our scope to try WEAVER. The first practitioner (C1) is building a pipeline for knowledge transfer, where they prompt an LLM to summarize content from transcripts into instructions. The second practitioner (C2) is building an IDE plugin to help developers understand unfamiliar code, developing LLM prompts to generate text summary for code segments. While these are two distinct scenarios, their shared challenge is that for practitioners working on novel tasks, it is often non-trivial to perform prompt engineering, especially because they do not have appropriate datasets for evaluating their prompts.

The case studies were IRB-approved and participants were compensated for their time.

**WEAVER supports quick and effective iterations.** Both C1 and C2 started with a seed concept and then refined the seed concept at least once based on their observations. For example, C2 first tried seed concepts *"challenges for summarizing a code script"* and *"reasons why people look for code summary"*, finding the recommended concepts generic and not their major concerns. They self-reflected through the process and identified the key scenario their plugin wants to support: (1) it targets novice programmers and (2) the most important application domain is data visualization. After this, they tried the seed concept *"specific challenges that novice programmers might have in comprehending data visualization code"* and found recommended concepts much more helpful.

**WEAVER helps practitioners find new model bugs by augmenting their existing analyses.** While we did not have participants rate each concept they explored, based on their think-aloud reflection, we note that they were able to find

many helpful concepts in a short amount of time. Even though practitioners have been working on their (LLM-backed) models for a while, they both obtained new insights into their models. First, WEAVER helped them observe issues they did not consider before. For example, in the seven concepts C1 tested, they found that the resulting instructions are always chronological even when there are detours in the input and steps reordering is desired. Second, WEAVER also helped them turn their prior, often fuzzy knowledge of problems or requirements into concrete testable concepts. For example, C1 turned their vague notion *"useful summaries should not take transcripts literal"* into concrete theories, including *"behind the transcript, there is a hidden thought process important for identifying key action steps."* Third, they were able to confirm model deficiencies they already suspected through systematic tests (e.g., *"transcript summaries are often too verbose"*). Similarly, C2 tested seven concepts and found *"different parameters for customization"* and *"when to use different data visualization APIs"* particularly novel and insightful.

Notably, while C1 used AdaTest for testing models on different concepts, C2 reused test cases from their existing datasets, showing WEAVER's flexibility with different test case creation techniques. That C2 still discovered new insights within their own dataset demonstrates WEAVER's capability for encouraging nuanced testing following requirements.

**WEAVER is useful beyond testing models after-the-fact.** While we mostly position WEAVER as a model testing tool, we find that its support for *requirements elicitation* supports the entire model development cycle (cf. the V-model, Fig. 1).

Although practitioners sometimes found it initially challenging to define seed concepts, they found the process itself valuable. For example, C2 eventually settled on *"specific challenges that novice programmers might have in comprehending [domain] code"*; they self-reflected how finding a good seed nudged them to state their goal explicitly *for the first time*. For them, this reflection happened too late to radically redesign their product, but it shows that WEAVER has the potential to support early-stage requirements engineering both for products and models. Meanwhile, C1 was inspired by concepts identified with WEAVER on model improvement. They experimented with different changes to prompts, encoding context for concepts they found challenging (e.g., step ordering).

## 6 Related Work

**Requirements elicitation.** Requirements engineering has been extensively studied (Van Lamsweerde, 2009). Despite many calls for the importance of requirements in ML (e.g., Rahimi et al., 2019; Vogelsang and Borg, 2019), requirements in ML projects are often poorly understood and documented (Nahar et al., 2022), which means that testers can rarely rely on existing requirements to guide their testing. Requirements elicitation is usually a manual and laborious process (e.g., interviews, focus groups, document analysis, prototyping), but the community has long been interested in automating parts of the process (Meth et al., 2013), e.g., by automatically extracting domain concepts from unstructured text (Shen and Breaux, 2022; Barzamini et al., 2022a). We rely on the insight that LLMs contain knowledge for many domains that can be extracted as KBs (Wang et al., 2020; Cohen et al., 2023), and apply this idea to requirements elicitation.

**Model evaluation, testing, and auditing.** Recent work on ML model evaluation (e.g., Ribeiro et al., 2020; Goel et al., 2021; Röttger et al., 2021; Yang et al., 2022) has pivoted from i.i.d. traditional accuracy evaluation to nuanced evaluation of model behaviors. As surveyed in our prior work (Yang et al., 2023), this line of research uses various test creation techniques, including slicing, perturbations, and template-based generation. While providing many useful tools, these approaches often assume an existing list of requirements and rarely engage with the question of *what to test.* Exiting research relied mostly on the knowledge of particular researchers, resulting in incomplete and biased requirements. For example, Ribeiro et al. (2020) explicitly state that their list of requirements in CheckList is not exhaustive and should be augmented by users with additional ones that are task-specific. Through LLM-assisted requirements elicitation, WEAVER helps users identify *what to test* systematically.

Various alternative methods have been proposed for identifying what to test. For example, error analysis (e.g., Naik et al., 2018; Wu et al., 2019a) and slice discovery (Eyuboglu et al., 2022) can help identify issues in existing datasets, but datasets are often incomplete and biased (Rogers, 2021), and can even be missing for emerging LLM applications where no dataset has been

pre-collected. Dataset-agnostic approaches like *adaptive testing* (Ribeiro and Lundberg, 2022; Gao et al., 2022) help users iteratively ideate concepts abstracted from generated test cases, but, as we confirmed, users tend to explore only *local* areas. These approaches engage in *bottom-up* style elicitation, which is reactive and may fare poorly with distribution shift. In contrast, WEAVER engages in *top-down* style elicitation, a more proactive process grounded in an understanding of the problem and domain.

Furthermore, algorithmic auditing (Metaxa et al., 2021) elicits concerns from people with different backgrounds, usually on fairness issues, to avoid being limited by the ideas of a single tester. However, it can be challenging to recruit, incentivize, and scaffold the auditors (Deng et al., 2023). In a way, WEAVER might complement such work by providing diverse requirements for individual testers or crowd auditors.

## 7  Discussion and Conclusion

In this work, we propose WEAVER, a tool that uses knowledge bases to guide model testing, helping testers consider requirements broadly. Thorough user studies and case studies, we show that WEAVER supports users to identify more, as well as more diverse concepts worth testing, can successfully mitigating users' biases, and can support real-world applications. Beyond being a useful testing tool, the underlying concept of WEAVER have interesting implications on ML model testing and development, which we detail below.

**Model testing in the era of LLMs.**  Throughout our user studies and case studies, we focused on testing "models" achieved by prompting LLMs. Here, we would like to highlight the importance of *requirements* in such cases. LLMs are increasingly deployed in different applications, and traditional model evaluations are becoming less indicative. With these models trained on massive web text, it is unclear what should be considered as "in-distribution evaluation data." Instead, the evaluation objectives heavily depend on what *practitioners* need, which should be reflected through well-documented requirements.

Meanwhile, as most practitioners are not NLP experts, they face challenges articulating how and what they should test about their prompted models (Zamfirescu-Pereira et al., 2023). As their use cases become more nuanced, it is also less likely for

them to find pre-existing collections on important concepts. As such, enabling each individual to identify *what-to-test* is essential. We hope WEAVER can be used for democratizing rigorous testing, just as LLMs democratized access to powerful models. Still, currently WEAVER relies purely on practitioners to identify requirements worth testing, which may result in mis-matched requirement granularity (cf. §3). Future work can explore more complex structures that can represent knowledge (e.g., from KBs to knowledge graphs), and advanced recommendation mechanisms for practitioners to find the best requirements to explore first.

**Rethinking requirements for ML development.** Though we position WEAVER to ground model testing in requirements, we expect it to be useful also in other development stages (cf. §5). For example, we expect that it can help developers think about high-level goals and success measures for their products (Rahimi et al., 2019; Passi and Barocas, 2019), to guide development early on. For example, building on the observation that requirement-based testing may help practitioners perform prompt engineering, we envision that future practitioners can use WEAVER for rapid prototyping, where they identify unique requirements, pair them with corresponding test cases, and achieve better overall performance either through ensembled prompts (Pitis et al., 2023) or prompt pipelines (Wu et al., 2022). Moreover, elicited model requirements themselves can serve as descriptions and documentation, which can foster collaboration and coordination in interdisciplinary teamwork (Nahar et al., 2022; Subramonyam et al., 2022). Notably, we believe WEAVER can support such iterations because it is built to be lightweight. In prior research, requirements engineering has sometimes been criticized to be too slow and bureaucratic, making developers less willing to dedicate time to this step. In contrast, WEAVER allows developers to easily adjust their exploration directions (through seeds and interactions), which makes it feasible to be integrated into more agile and iterative development of ML products where requirements are evolving quickly.

## Limitations

**Availability of domain knowledge in LLMs.** LLMs encode a vast amount of knowledge, but may not include very domain-specific knowledge for specialized tasks, very new tasks, or tasks where

relevant information is confidential. Our technical implementation fundamentally relies on extracting knowledge from LLMs and will provide subpar guidance if the model has not captured relevant domain knowledge. Conceptually our approach to guide testing with domain knowledge would also work with other sources of the knowledge base, whether manually created, extracted from a text corpus (Shen and Breaux, 2022; Barzamini et al., 2022a), or crowdsourced (Metaxa et al., 2021).

**Impacts from biases in LLMs.**  WEAVER uses LLMs to build knowledge bases such that users can elicit diverse requirements. However, LLMs themselves are found to be biased, sometimes untruthful, and can cause harm (Nadeem et al., 2021; Kumar et al., 2023). Therefore, users should carefully interpret results from WEAVER in high-stake applications.

**Threats to validity in human-subject evaluations.**  Every study design has tradeoffs and limitations. In our evaluation, we intentionally combined multiple different kinds of user studies to triangulate results.

First, we conducted a user study as a controlled experiment. While the results are very specific and created in somewhat artificial settings and must be generalized with care (limited external validity), the study design can enact a high level of control to ensure high confidence in the reliability of the findings in the given context with statistical techniques (high internal validity). For example, regarding external validity, results may not generalize easily to other tasks that require different amounts of domain understanding or are differently supported by the chosen test case creation technique, and our participant population drawn from graduate students with a technical background may not equally generalize to all ML practitioners. There are also some threats to internal validity that remain, for example, despite careful control for ordering and learning effects with a Latin square design and assuring that the four groups were balanced in experience ('years of ML experience' and 'NLP expertise' asked in the recruitment survey before assignment), we cannot control for all possible confounding factors such as prior domain knowledge, gender, and motivation. In addition, we rely on clustering and similarity measures among concepts for our dependent variables, which build on well-established concepts but may not always align with individual subjective judgment.

Second, we conducted case studies in real-world settings with practitioners (high external validity) but can naturally not control the setting or conduct repeated independent observations (limited internal validity). With only two case studies, generalizations must be made with care.

This tradeoff of external and internal validity is well understood (Siegmund et al., 2015). Conducting both forms of studies allows us to perform some limited form of triangulation, increasing confidence as we see similar positive results regarding WEAVER's usefulness for discovering diverse concepts.

**Subjectivity in human judgments.**  All model testing requires judgment whether a model's prediction for a test example is correct. We noticed that user study participants and sometimes also case study practitioners struggled with determining whether model output for a specific test example was a problem, and multiple raters may sometimes disagree. For our purposes, we assume it is the tester's responsibility of identifying which model outputs they consider problematic, and we do not question any provided labels. This, however, reminds us that like data annotation (Santy et al., 2023), any model testing process will likely bring in testers' biases, as they get to decide what is right and what is wrong. In practice, a broader discussion among multiple stakeholders may be required to identify what model behavior is actually expected and a decomposition of model testing using requirements might be helpful to foster such engagement.

## Ethics Statement

**Research Reproducibility.**  While our experiments are mostly conducted on the closed API (text-davinci-003) provided by OpenAI, none of the conceptual contributions of our paper relate to specific models or APIs. The concrete evaluation results depend on how humans interact with specific models, but the approach can be used with other models. Indeed, our extra experiments with llama-2-13b-chat on the climate change task show that the generated concepts from open-source models achieve substantial levels of recall (83% vs. 91% originally). This supports that the idea behind WEAVER is reproducible. While users may see somewhat different KBs through different runs of the same/different LLMs, they get similar chances of seeing useful concepts, receive a similar level of

support on requirement elicitation (our core contribution), and will be able to yield similar model testing effectiveness.

**Human-subject Experiments.** Our studies had been approved by our IRB before it was conducted, as is standard practice for human-subject experiments. We recruited all participants through emails, and all of them are graduate students with varying ML/NLP experience (see details in Appendix D.1). The participants were compensated for their time ($20 per hour). As part of testing, they may write or review text that is abusive, dangerous, hateful, or offensive—they were made aware of this fact and could end participation at any time.

## Acknowledgements

This work was supported in part by the National Science Foundation (#2131477) and gift funds from Adobe, Oracle, and Google. Work by Brower-Sinning and Lewis was funded and supported by the Department of Defense under Contract No. FA8702-15-D-0002 with Carnegie Mellon University for the operation of the Software Engineering Institute, a federally funded research and development center (DM23-2019).

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

## A   Complete List of Used Prompts

```
# Prompts for expanding knowledge bases

{context}
{list_prompt} Pay attention to the context above.
    Summarize in a JSON list.

'''json

## Example list prompts
TYPEOF: List {N} types of {concept}.
PARTOF: List {N} parts or aspects of {concept}.
HASPROPERTY: List {N} descriptions of {concept}.
USEDFOR: List {N} things {concept} could be used
    for.
ATLOCATION: List {N} locations {concept} could
    appear in.
CAUSES: List {N} consequences of {concept}.
MOTIVATEDBY: List {N} motivations behind {
    concept}.
OBSTRUCTEDBY: List {N} things, entities, or
    people against {concept}.
MANNEROF: List {N} ways to do {concept}.
LOCATEDNEAR: List {N} things that often locates
    near {concept}.
CAPABLEOF: List {N} things that {concept} is
    capable of.
HASSUBEVENT: List {N} subevents of {concept}.
HASPREREQUISITE: List {N} things that happen
    before {concept}.
DESIRES: List {N} things that {concept} desires.
CREATEDBY: List {N} creators of {concept}.
SYMBOLOF: List {N} symbols of {concept}.
CAUSESDESIRE: List {N} desires caused by {
    concept}.
MADEOF: List {N} materials of {concept}.
RECEIVESACTION: List {N} actions that can be
    done to {concept}.
DESIREDBY: List {N} entities or people that
    desire {concept}.
CREATES: List {N} things that {concept} creates.
CAUSEDBY: List {N} things that cause {concept}.
DONEBY: List {N} entities or people that can do
    {concept}.
DESIRECAUSEDBY: List {N} things that cause
    desire of {concept}.
DONETO: List {N} entities or people that {
    concept} can be done to.
RELATEDTO: List {N} concepts related to {concept
    }.

## Example contexts prompts
TYPEOF: {concept} is a type of {parent_concept}.
PARTOF: {concept} is a part of {parent_concept}.
HASPROPERTY: {parent_concept} is described as {
    concept}.
USEDFOR: {parent_concept} is used for {concept}.
ATLOCATION: {parent_concept} locates at {concept
    }.
CAUSES: {parent_concept} causes {concept}.
MOTIVATEDBY: {parent_concept} is motivated by {
    concept}.
OBSTRUCTEDBY: {parent_concept} is obstructed by
    {concept}.
MANNEROF: {concept} is a way to do {
    parent_concept}.
LOCATEDNEAR: {concept} locates near {
    parent_concept}.
CAPABLEOF: {parent_concept} is capable of {
    concept}.
HASSUBEVENT: {concept} happens during {
    parent_concept}.
HASPREREQUISITE: {concept} happens before {
    parent_concept}.
DESIRES: {parent_concept} desires {concept}.
CREATEDBY: {concept} creates {parent_concept}.
SYMBOLOF: {concept} is a symbol of {
    parent_concept}.
CAUSESDESIRE: {parent_concept} causes desire of
    {concept}.
MADEOF: {parent_concept} is made of {concept}.
RECEIVESACTION: {parent_concept} receives action
     of {concept}.
DESIREDBY: {concept} desires {parent_concept}.
CREATES: {parent_concept} creates {concept}.
CAUSEDBY: {concept} causes {parent_concept}.
DONEBY: {concept} does {parent_concept}.
DESIRECAUSEDBY: {concept} causes desire of {
    parent_concept}.
DONETO: {parent_concept} is done to {concept}.
RELATEDTO: {concept} is related to {
    parent_concept}.
```

```
# Prompts for the tested models in the user
    study

# Stance detection on feminism
Carefully classify a sentence's stance on
    feminism. The labels are \"favor\", \"
    against\" or \"none\". Only reply with the
    label.
Sentence: {example}

# Stance detection on climate change
Carefully classify a sentence's stance on
    combating climate change. The labels are \"
    favor\", \"against\" or \"none\". Only reply
     with the label.
Sentence: {example}
```

The path contexts in our full prompt are used to mitigate the issue of polysemy and deviation when users explore deeper in the graph.

---

SURVEY QUESTIONS
- Rate whether you agree with the statement: *Concept knowledge graph helps me find more diverse model bugs*. Write a justification for your rating.
- Rate whether you agree with the statement: *Concept knowledge graph helps me find more important model bugs*. Write a justification for your rating.
- Rate whether you agree with the statement: *Concept knowledge graph helps me test the model more holistically*. Write a justification for your rating.
- Rate whether you agree with the statement: *I want to use the tool with knowledge graph to test the model I build/use in the future.*. Write a justification for your rating.
- If you want to use the tool in the future, what is the model and task you want to use it for?
- Feedback on how to improve the tool in the future.

Figure 8: Post-study survey questions.

## B   User Interface

In Figure 9, we show the complete user interface.

## C   Task Descriptions

*Hateful meme detection.* Hateful meme detection (Kiela et al., 2020) requires classifying an image (meme with text) as hateful or non-hateful. This task is challenging in that it requires multi-modal reasoning in order to classify the original meme and its confounders correctly

*Pedestrian detection.* Pedestrian detection (Zhang et al., 2017) requires detecting and localizing pedestrians in images. Though it is one of the longest-standing problems in computer vision, people still observe generalizability issues in existing detectors (Hasan et al., 2021).

*Stance detection.* Stance detection requires classifying texts as either being in favor of, against, or neutral toward the given target (Mohammad et al., 2016). The task is crucial for understanding the public's perception of given targets We selected two targets for our evaluation: *feminism* and *climate change*, which are previously explored for Tweets (Mohammad et al., 2016).

## D   User Study

### D.1   Study Design Details

**Participants.**   We recruited 20 participants (graduate students with varying ML/NLP experience). Among them, 70% have worked on ML for more than three years; 80% rate themselves as at least somewhat familiar with NLP; 60% are working on a project using NLP at the time of the study. We randomly assigned participants to the four experimental groups. The four groups are comparable in terms of ML/NLP experience and familiarity. The participants were compensated $20 per hour.

### D.2   Post-study Survey

We share our survey questions (Figure 8) and users' responses (Table 2).

### D.3   Quantitative Analysis

We show the ANOVA analysis results in Table 3.

## E   Additional Data on Running WEAVER

It takes 30 seconds on average to generate a default KB from scratch (with around 500 concepts) on our machine (Precision 3650 workstation, with Intel(R) Xeon(R) W-1350 CPU and 32GB memory). When users explore the KB interactively and expand a node, the query takes 8 seconds on average. The wait can be greatly reduced via pre-fetching (i.e., expanding the node on display in the background before the actual query), which has been implemented in WEAVER.

## F   WEAVER with Open-source LLMs

We conducted an extra experiment to evaluate whether open-source LLMs can also generate comprehensive KBs. We generated another KB on the climate change task, using `llama-2-13b-chat` with 4-bit quantization. We found that the generated KB achieved comparable levels of recall (83% vs. 91% with `text-davinci-003`).

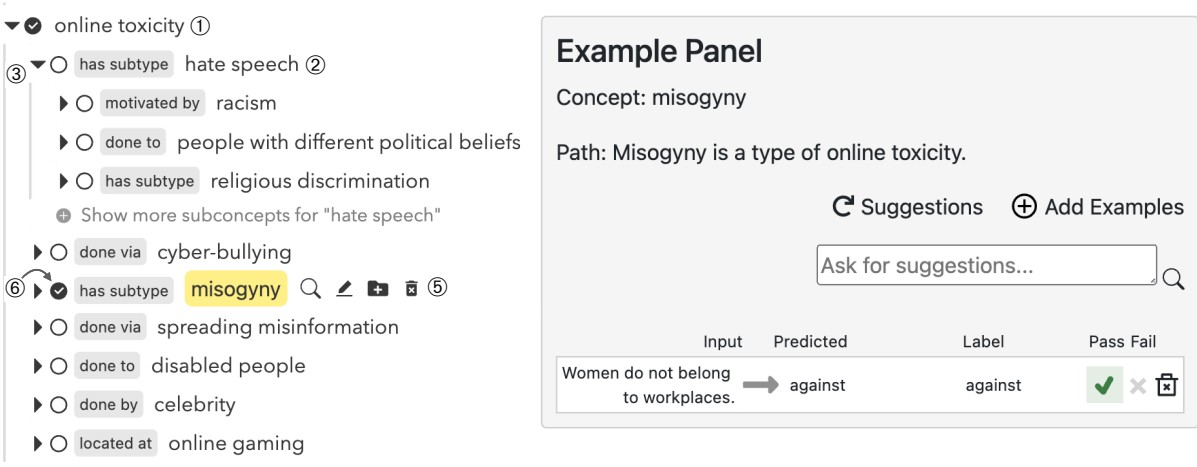

Figure 9: Complete user interface for WEAVER. On the right is our re-implemented version of AdaTest to assist users to create test cases.

| | Statement | Distribution |
|---|---|---|
| Q1 | Knowledge base helps me find more diverse model bugs. | *75%* ▬▬▬▬▬▬ *10%* |
| Q2 | Knowledge base helps me find more important model bugs. | *45%* ▬▬▬▬▬▬ *15%* |
| Q3 | Knowledge base helps me test the model more holistically. | *80%* ▬▬▬▬▬▬ *0%* |
| Q4 | I want to use WEAVER to test the model I build/use in the future. | *95%* ▬▬▬▬▬▬ *5%* |

■ Strongly agree ■ Agree ▨ Neutral ■ Disagree ■ Strongly disagree

Table 2: Participants' responses in the post-study survey.

| **Number of found concepts** | **ges** |
|---|---|
| Interv.: Used WEAVER? | 0.307** |
| Order: Tool in first task? | 0.008 |
| Task number | 0.000 |
| **Number of hit clusters** | **ges** |
| Interv.: Used WEAVER? | 0.408** |
| Order: Tool in first task? | 0.039 |
| Task number | 0.006 |

$^{**}p < 0.01$        $N = 40$

Table 3: User study ANOVA analysis results. We measure effect size with ges (generalized Eta-Squared measure of effect size).