# OpenReview forum: "Beyond Testers’ Biases: Guiding Model Testing with Knowledge Bases using LLMs"
_EMNLP/2023/Conference — EMNLP 2023 Findings_

### Official Review · Reviewer_Mdj9 · 2023-08-02

**Soundness:** 4

**Ethical Concerns:**

Yes

**Excitement:**

4: Strong: This paper deepens the understanding of some phenomenon or lowers the barriers to an existing research direction.

**Justification For Ethical Concerns:**

- I do not see if an IRB approves your user study and how you compensate for your users (I have scoured the manuscript and the supplementary materials section on OpenReview, but I could not find anything). This approval and subsequent discussions are essential as some of the datasets you use (e.g., hateful memes and stance detection for feminism) could impose harm on the testers.

**Paper Topic And Main Contributions:**

This paper proposes a tool (Weaver) to help NLP model testers automatically develop specifications under an initial abstract and less well-defined topic, such as "online toxicity." The tool addresses the limitations of a previous tool called AdaTest by mitigating the burden of testers of summarizing and organizing LM-generated test cases into a test tree. The controlled experiments show that Weaver can help testers explore more diverse topics, which benefits NLP testing.

**Questions For The Authors:**

1. In software engineering, the requirement is specified before the developers even write the software, but the authors are testing existing models. Do authors believe the way of requirement engineering is different for traditional software and machine learning models?

2. The Weaver tries to build and recommend fine-grained topics based on a knowledge graph (KG). However, there are a few issues in this process:
   1. In Line 160, the generated topics are not checked for their validity.
   2. The authors use the relations provided by ConecptNet as-is without considering its bias (for example, its coverage may be insufficient). Even though this choice seems reasonable, I expect the authors to discuss the bias of using existing relations in ConceptNet.
   3. In Line 199, the authors propose to maximize diversity and relevance at the same time. However, it would be better to formulate this as a multi-objective optimization problem since we will likely land on a solution with very high diversity but low relevance.
3. Is your user study single-blind or even double-blind? I do not see whether the users mentioned in Section 4 know which group (control vs. treatment) they are in, and I don't know if the authors could unconsciously influence the users.
4. The design of Figure 5 is quite interesting. However, I do not understand the role of gray lines.
5. Are the distances between concepts in Figure 6 drawn to scale? How do the authors place the clusters? These plotting details are important as the authors are trying to use this graphic as part of the evidence that their proposed approach (i.e., orange clusters) are better than the control group (i.e., blue clusters) as it generates clusters that are more spread out.
6. Missing minor details
   1. L327 - L330: What exact models do you use?
   2. L371 - L386: What are the configurations of your hierarchical clustering?

**Reasons To Accept:**

- The paper is trying to solve a real and useful problem - requirement elicitation for NLP models; it is also one of the first few papers trying to do so, making it timely and relevant to the NLP testing community.
- The authors' method is sound overall, and their tool seems user-friendly, although the authors have not released a demo version for reviewers to try.

**Reasons To Reject:**

- This paper tries to improve one component of the previous AdaTest system, which looks incremental. Furthermore, the author's approach does not improve the error rates (as the authors acknowledge in L405 - L408) of the tested models, even though it helps explore more topics.

- The authors rely on the closed-source API - `text-davinci-003` from OpenAI to generate the specific list of topics under an abstract topic; the system the authors try to improve (i.e., AdaTest) also uses `text-davinci-003` or similar APIs. However, this service is going to discontinue in January 2024. I expect to see authors rebuild their system (including both Weaver and AdaTest) on open-source models and reproduce the experiments; the proposed approach should show that it can work with and without a strong base LM (i.e., text-davinci-003).

  As a side note, the authors claim to use ChatGPT (`gpt-turbo-3.5`) throughout the text, but they use text-davinci-003. It will also be nice to see authors reproduce results on `gpt-turbo-3.5` to make the experiment results and claims in the paper match each other.

- The authors' user study seems incomplete from what I can see on L351 - L355. The authors may want to spend more time on the user study to make the experiments more complete before making a submission.

**Reproducibility:**

1: Could not reproduce the results here no matter how hard they tried.

**Reviewer Confidence:**

5: Positive that my evaluation is correct. I read the paper very carefully and I am very familiar with related work.

**Typos Grammar Style And Presentation Improvements:**

- Table 1: The `# Conecpt` should be average rather than sum.

- L370: Redundant "to".

---

> ### Author Rebuttal · Authors · 2023-08-29
>
> Thank you for acknowledging the value of requirement elicitation for NLP model testing! We do plan to open source both the Weaver KB builder and user interface upon paper acceptance.
>
> **Contribution**
>
> We would like to emphasize that our core contribution is the **concept and method of requirements engineering to improve testing**. It is purely out of convenience that we used AdaTest as our primary test generator, but Weaver and AdaTest make contributions on completely different parts of the larger problems, i.e., what to test vs. how to test. As mentioned in our case study, AdaTest is not necessarily the only test generator that can be used. Weaver’s interface can be paired with other test generation methods as well.
>
>
> **Dependence on closed-source API**
>
> We acknowledge that we use a closed-source API (`text-davinci-003`) for building the KB in Weaver. To see whether we can use an open-source model alternative, we performed an extra experiment with `llama-2-13b-chat` on the climate change task (in addition to what has been conducted in the paper). We found that the generated concepts from open-source models achieve comparable levels of recall (82.7\% vs. 91.4\% originally). Again, we want to emphasize that none of the conceptual contributions of this paper relate to specific models or APIs. The model is important for creating the KB but multiple papers have now robustly shown that this is possible using different models (`text-davinci-002`, gpt2, BERT) [1, 2], so we expect this to replicate conceptually even when an exact reproduction of data may not be possible.
>
> > “As a side note, the authors claim to use ChatGPT (`gpt-turbo-3.5`) throughout the text, but they use `text-davinci-003`.”
>
> To clarify, we did evaluate Weaver on ChatGPT (`gpt-turbo-3.5`) in our user study, while we built Weaver using `text-davinci-003`, the most powerful LLM API accessible at the time of our system development.
>
>
> **User study design**
>
> > User study completeness
>
> The reviewer seems to misinterpret reports from a pilot study in L351 - L355 as an indication of an incomplete study. In fact, a pilot study is a common approach in human-subject studies to validate and iterate on study designs. All the results were based on observations in the final study.
>
> Here, we illustrate more on our rationale for *designing the study to be 25 min model testing + 5 min additional exploration, instead of making the study longer*, according to our pilot study observations: The primary goal of our user study is to see whether Weaver *better supports the concept exploration step in the context of model testing* – In other words, we are more interested in seeing what concepts would catch participants’ attention, and less interested in seeing them create test cases for every concept. In the *pilot study*, we noticed the latter (writing test cases) takes much longer time and effort – completely testing all the concepts users find interesting requires ~2 hours. To ensure that participants can focus on the primary goal, we chose to only have them test a subset of concepts to verify the error rate per concept and get grounded in what concepts are worth testing, such that they can spend a larger proportion of time collecting concepts to be tested later.
>
> > Single / double-blindness of our study
>
> The discussion of single-blind vs. double-blind does not apply to our within-subject study design. All participants perform tasks in the control and the experimental conditions (the order and tasks are intentionally varied and counterbalanced between participants). It is obvious to participants that they perform the task in different conditions, though they are not informed which condition is the treatment condition. The experimenter also can identify the conditions. Within-subject designs are very common for controlled experiments and have tradeoffs with between-subject designs (where double-blind studies are possible) [3]. In our context, we picked within-subject for reducing noise while keeping the number of participants manageable.
>
>
> **Is requirements engineering different for traditional software and ML models?**
>
> That’s a very thoughtful question! As argued in our paper, the *state of practice* between requirements engineering in traditional software and ML models is certainly very different. More importantly, we also expect there to be substantial differences *conceptually* due to a lack of a clear specification for ML models: In traditional software, requirements engineers create the specifications. In machine learning, we expect that requirements are discovered more incrementally during testing rather than specified upfront.
>
> Further, to clarify, the V-model emphasizes the mapping between requirements and tests, but the process does not necessarily need to be sequential.
>
>
> **The validity and creation of KB**
>
> > Invalid Concepts in the KB
>
> We did not filter out invalid concepts as we empirically observed that concepts suggested by the LLM are mostly valid. In practice, users can always delete invalid concepts or simply ignore them. However, we did not observe users deleting invalid concepts or making relevant complaints in the post-study survey. In an extra experiment to evaluate precision, we sampled 200 edges from our 4 KBs evaluated in Section 3, 50 each, and manually judge their correctness. We found 161 out of 200 edges valid, achieving a precision of 80.5\%.
>
> > Dependence on ConceptNet
>
> We picked ConceptNet because it's more about common sense and has generic semantic relations between concepts. Though knowledge tends to have bias, we expect generic semantic relations to be more or less generalizable – an assumption that’s validated by our user study and case studies. We will discuss alternative KBs in the revision, such as WikiData, DBpedia, etc., that tend to focus on more specific types of semantic relations.
>
>
>
> **Other technical details**
>
> > What is the role of gray lines in Figure 5?
>
> The grey line is the identical line that plots the ideal situation, where each identified concept is distinct and falls into a different cluster.
>
> > Are the distances between concepts in Figure 6 drawn to scale? How do the authors place the clusters?
>
> Yes. We use t-SNE to project all concepts’ SentenceBERT embeddings into a 2d space. We then filter out concepts explored by other participants and draw the clusters such that their relative positions are minimally changed.
>
> > What exact models do you use?
>
> We started from the open-source huggingface models (https://huggingface.co/cardiffnlp/twitter-roberta-base-stance-climate, https://huggingface.co/cardiffnlp/twitter-roberta-base-stance-feminist) fine-tuned on the TweetEval dataset. We ended up just testing `gpt-3.5-turbo` as it achieves better performance.
>
> > What are the configurations of your hierarchical clustering?
>
> We use the following setting: `sklearn.cluster.AgglomerativeClustering(n_clusters=None, affinity=’cosine’, memory=None, connectivity=None, compute_full_tree='auto', linkage='average', distance_threshold=0.7, compute_distances=False)`
>
>
> **References**
>
> [1] Chenguang Wang, Xiao Liu, and Dawn Song. 2020. Language models are open knowledge graphs. arXiv preprint arXiv:2010.11967.
>
> [2] Roi Cohen, Mor Geva, Jonathan Berant, and Amir Globerson. 2023. Crawling the internal knowledge base of language models.
>
> [3] George E. P. Box. 2009. Statistics for experimenters: design, innovation, and discovery. Wiley-Blackwell.

---

### Official Review · Reviewer_hoyj · 2023-08-03

**Soundness:** 3

**Excitement:**

2: Mediocre: This paper makes marginal contributions (vs non-contemporaneous work), so I would rather not see it in the conference.

**Paper Topic And Main Contributions:**

The authors propose a system for suggesting different concepts to be evaluated related to the same task. The concepts are suggested using KB and shown using a tree structure. The KB is automatically created using the knowledge contained in LLMs.
The main contributions are the system presented for suggesting additional tests. I think the main contribution is clear from the following excerpt in the paper "However, prior work has focused on how to write test cases, not what tests to write."

**Questions For The Authors:**

A: Besides recall, have you evaluated the concepts generated by weaver that are not present in the ground-truth concepts? I mean, some kind of evaluation taking into account precision. This is important because these additional concepts could mislead users

**Reasons To Accept:**

- The method for creating the KB using LLMs and show it to the users.

**Reasons To Reject:**

- Although the concepts suggested are important for a developer/researcher, I miss how this work is connected to traditional benchmarking of systems for improving testing or deeper analysis. The proposed system can be connected to a test generator but I see the proposed interface as a recommender system of some concepts instead of a tool for a quick testing of error analysis, or something similar, in new developments.


**Reproducibility:**

4: Could mostly reproduce the results, but there may be some variation because of sample variance or minor variations in their interpretation of the protocol or method.

**Reviewer Confidence:**

5: Positive that my evaluation is correct. I read the paper very carefully and I am very familiar with related work.

**Typos Grammar Style And Presentation Improvements:**

- Line 370, "To to" -> "To"

---

> ### Author Rebuttal · Authors · 2023-08-29
>
> **Relation to traditional benchmarking and model testing approaches**
>
> Thank you for acknowledging that the concepts Weaver suggests are important! Better contextualizing Weaver based on existing model analysis literature is indeed important, and we will add these discussions to the paper.
>
> Similar to existing literature on SE-inspired NLP model testing, our work is *complementary to traditional benchmarking*, in that it offers more fine-grained information about concrete model behaviors beyond standard metrics. The testing schema is becoming more important as LLM+prompting is applied to various novel tasks where traditional benchmarking is nearly impossible due to the lack of datasets (as in our case studies).
>
> Among the related work in model testing (CheckList, AdaTest), as we have reviewed in Section 6 and as the reviewer has noticed, “prior work has focused on how to write test cases, not what tests to write.” In contrast, the core contribution of Weaver is *on systematically revealing “what to test”*, which precisely makes it “a recommender system of *diverse but relevant* concepts” – this is the key novelty of our approach over the state of the art. Similarly, we note that the goal of Weaver is not “quick testing / error analysis,” but more towards supporting *thorough* testing of *all relevant requirements* for a task even though it might not make testing itself faster. We will clarify that Weaver’s testing power comes from combining Weaver’s LLM+ConceptNet-based KB with existing test generators like AdaTest.
>
>
> **Precision vs. recall in Weaver concept evaluation**
>
> Thank you for the suggestion. We carefully considered how to evaluate the quality of presented concepts, but are limited by available ground truth data. As described in Section 3, we have two sources of *partial* ground truth: dataset analysis and user study in prior papers, and concepts identified in our user study. Both sources are incomplete. For example, the Hateful Meme Detection ground truth is collected from a user study, where users iterate over their models and find important concepts. As a result, these ground truths only reflect concepts that are in-distribution and are known to be issues, but do not contain other relevant concepts that would be important for practical model deployment. With this information, we can only compute recall based on known concepts, but cannot compute precision.
>
> We did perform a follow-up, manual validity check to evaluate precision. Following prior work [1], we sampled 200 edges from our 4 KBs evaluated in Section 3, 50 each, and manually judged their correctness. We found 161 out of 200 edges valid, achieving a precision of 80.5\%. We will add these justifications and results to the paper revision.
>
> As a proxy for practical usefulness (i.e., practically acceptable precision), consider that participants in our studies could navigate the KG without being overwhelmed by useless proposed concepts. Indeed, we did not observe users deleting invalid concepts or making relevant complaints in the post-study survey.
>
>
> **References**
>
> [1] Roi Cohen, Mor Geva, Jonathan Berant, and Amir Globerson. 2023. Crawling the internal knowledge base of language models.

---

### Official Review · Reviewer_rtMG · 2023-08-05

**Soundness:** 3

**Excitement:**

3: Ambivalent: It has merits (e.g., it reports state-of-the-art results, the idea is nice), but there are key weaknesses (e.g., it describes incremental work), and it can significantly benefit from another round of revision. However, I won't object to accepting it if my co-reviewers champion it.

**Paper Topic And Main Contributions:**

The paper introduces an interactive tool, WEAVER, for guiding model testing. WEAVER uses large language models to generate concepts by prompting with pre-defined relations from ConceptNet. It applies a greedy algorithm to select the concepts that are diverse but still relevant to the seed queries. Then the generated concepts are expected to help developers or researchers to come up with better testing examples. The paper presents a user study on two tasks, stance detection for feminism and stance detection for climate change. The experiments show that both NLP experts and non-experts identified more, as well as more diverse concepts worth testing when using WEAVER. The paper also provides two case studies to demonstrate how WEAVER can help practitioners test models in real-world applications.


**Reasons To Accept:**

1. The paper proposes that it is important to identify "what to test" in model testing. They introduced an interactive testing tool that uses large language models to generate concepts for model testing.

2. The user study and case studies provide empirical evidence of the effectiveness of the tool to guide model testing. The results show that human users identified more, as well as more diverse concepts worth testing when using WEAVER.


**Reasons To Reject:**

1. The paper is a bit difficult to follow. It lacks some details on the technical implementations. It would be beneficial for readers if the paper provided more descriptions, e.g., in Section 2.2, does each query requires building a new graph (involving SentenceBERT and GPT-2)? How long does this process take? Does the user need to wait for this each time the seed query is changed? In Section 3, line 262, what are the 144 ground-truth concepts?


2. The case studies in Section 5 are insightful but too brief. Readers can benefit from more details, e.g., what are the practical problems, what are the seed queries practitioners use, what concepts are most helpful, etc?




**Reproducibility:**

2: Would be hard pressed to reproduce the results. The contribution depends on data that are simply not available outside the author's institution or consortium; not enough details are provided.

**Reviewer Confidence:**

4: Quite sure. I tried to check the important points carefully. It's unlikely, though conceivable, that I missed something that should affect my ratings.

---

> ### Author Rebuttal · Authors · 2023-08-29
>
> We thank the reviewer for recognizing our contribution to the model testing procedure!
>
> **Technical implementation details**
>
> 1. Indeed, Weaver will need to rebuild the KB each time a new seed is queried, which is a fundamental strength supporting Weaver’s flexibility and generalizability. Generating a default KB from scratch (with ~500 concepts) takes ~30 seconds on our machine (Precision 3650 workstation, with Intel(R) Xeon(R) W-1350 CPU and 32GB memory). When users explore the KB interactively and expand a node, the query takes ~8 seconds. The wait can be greatly reduced via pre-fetching (i.e., expanding the node on display in the background before the actual query), which has been implemented in Weaver. In practice (in our case studies), users did not complain about the usability of Weaver.
>
> 2. The ground-truth concepts in Section 3 are collected from previous work and user studies (L251-259), we shared the full list of the ground-truth concepts in the link here: https://figshare.com/s/481a69fa1b36dbd76088
>
> We plan to extend our write-up on the technical implementation upon acceptance, and open source both the Weaver KB builder and user interface.
>
> **Case study details**
>
> We are glad that the reviewer found the case study insightful! We are happy to add more concrete evidence on practitioner usage.
>
> 1. *Practical problem*: As described in Section 5, C1 is building a pipeline for knowledge transfer, where they prompt an LLM to summarize content from transcripts into instructions. C2 is building an IDE plugin to help developers understand unfamiliar code, developing LLM prompts to generate text summaries for code segments. While these are two distinct scenarios, their shared challenge is that for practitioners working on novel tasks, it is often non-trivial to perform prompt engineering, especially because they don’t have appropriate datasets for evaluating their prompts.
>
> 2. *Concrete exploration procedure*: Both C1 and C2 started with a seed concept of their own wording, then refined the seed concept at least once based on their observations. For example, C2 first tried seed concepts “*challenges for summarizing a code script*” and “*reasons why people look for code summary*”, finding the recommended concepts generic and not their major concerns. They self-reflected through the process and identified the key scenario their plugin wants to support: 1) it targets novice programmers and 2) the most important application domain is data visualization. After this, they tried the seed concept “*specific challenges that novice programmers might have in comprehending data visualization code*” and found many helpful concepts.
>
> 3. *Useful concepts*: While we did not have participants rate each concept they explored, based on their think-aloud reflection, we note that they were able to find many helpful concepts in a short amount of time. At a high level, concrete concepts that help practitioners explicate existing requirements or discover new requirements tend to be more helpful. For example, C1 tested 7 concepts and they (a) re-confirmed their concerns by finding multiple examples where the summaries are too verbose and (b) found several other issues by testing concepts “*right tone and voice for audience*” and “*understanding of thought process*”. C2 tested 7 concepts and found “*different parameters for customization*” and “*when to use different data visualization APIs*” particularly helpful.

---

### Meta-Review · Area_Chair_Yz1V · 2023-09-19

**Recommendation:** 3

**Metareview:**

This paper proposes a new interactive system, WEAVER, to help/guide NLP model testers. The system generates concepts from a large language model (LLM) that are diverse but relevant to a task. The idea is that the generated concepts can provide additional tests for modeling testing. The authors' experiments show that WEAVER can help testers explore similar topics.

[Paper Clarity]: Both reviewers rtMG and Mdj9 raise concerns about the paper's clarity. In particular, reviewers rtMG have issues with the lack of technical detail around how the proposed system works. Whereas reviewer Mdj9 concerns are with regard to details not mentioned in the paper around the validity of new concepts, the authors's choice to use ConceptNet, and the authors' optimization objective.

[Traditional Software Engineering Testing]:
Both reviewers hoyj and Mdj9 raise concerns about how the proposed approach is related to traditional software engineering testing that provides quick testing/error analysis.

---

### Decision · Program_Chairs · 2023-10-07

**Decision:**

Accept-Findings

**Comment:**

This paper proposes a new interactive system, WEAVER, to help/guide NLP model testers. The system generates concepts from a large language model (LLM) that are diverse but relevant to a task. The idea is that the generated concepts can provide additional tests for modeling testing. The authors' experiments show that WEAVER can help testers explore similar topics.

[Paper Clarity]: Both reviewers rtMG and Mdj9 raise concerns about the paper's clarity. In particular, reviewers rtMG have issues with the lack of technical detail around how the proposed system works. Whereas reviewer Mdj9 concerns are with regard to details not mentioned in the paper around the validity of new concepts, the authors's choice to use ConceptNet, and the authors' optimization objective.

[Traditional Software Engineering Testing]:
Both reviewers hoyj and Mdj9 raise concerns about how the proposed approach is related to traditional software engineering testing that provides quick testing/error analysis.